# Smooth Fictitious Play in Stochastic Games with Perturbed Payoffs and Unknown Transitions

**Lucas Baudin**
Université Paris-Dauphine - PSL
`lucas.baudin@dauphine.eu`

**Rida Laraki**
CNRS, Université Paris-Dauphine - PSL
University of Liverpool

## Abstract

Recent extensions to dynamic games (Leslie et al. [2020], Sayin et al. [2020], Baudin and Laraki [2022]) of the well-known fictitious play learning procedure in static games were proved to globally converge to stationary Nash equilibria in two important classes of dynamic games (zero-sum and identical-interest discounted stochastic games). However, those decentralized algorithms need the players to know exactly the model (the transition probabilities and their payoffs at every stage). To overcome these strong assumptions, our paper introduces regularizations of the systems in Leslie et al. [2020], Baudin and Laraki [2022] to construct a family of new decentralized learning algorithms which are model-free (players don't know the transitions and their payoffs are perturbed at every stage). Our procedures can be seen as extensions to stochastic games of the classical smooth fictitious play learning procedures in static games (where the players best responses are regularized, thanks to a smooth strictly concave perturbation of their payoff functions). We prove the convergence of our family of procedures to stationary regularized Nash equilibria in zero-sum and identical-interest discounted stochastic games. The proof uses the continuous smooth best-response dynamics counterparts, and stochastic approximation methods. When there is only one player, our problem is an instance of Reinforcement Learning and our procedures are proved to globally converge to the optimal stationary policy of the regularized MDP. In that sense, they can be seen as an alternative to the well known Q-learning procedure.

## 1 Introduction

Fictitious play (FP) (Brown [1951], Robinson [1951]) is the oldest and most famous independent (i.e., decentralized) learning algorithm in game theory. It is a simple behavioural strategy that agents may use to repeatedly play a fixed normal form game $G$. Namely, at every repetition of $G$, each player best replies to the empirical distribution of the uncorrelated past actions of the opponents.

When $G$ is a finite game, it has been shown that the empirical distribution of the players' actions under FP globally converges to the set of mixed Nash equilibria, when $G$ is zero-sum (Brown [1951], Robinson [1951]) or identical-interest (Monderer and Shapley [1996]). The FP procedure was extensively studied since then with numerous additional convergence results (Miyasawa [1961], Berger [2005], van Strien and Sparrow [2011]) and a number of generalizations (Leslie and Collins [2006], Benaïm and Faure [2013] to cite a few).

Unfortunately, FP has no guarantee in terms of regret. Indeed, it may well lead to the worst possible payoff in a zero-sum game if the opponent best-replies to it at every stage. Fortunately, the smooth version of FP (call it SFP), also named stochastic or regularized fictitious play, is proved to have the no-regret property (Fudenberg and Levine [1995], Benaïm and Faure [2013]) and to converge to regularized Nash Equilibria in the classes mentioned so far (Hofbauer and Sandholm [2002], Berger [2005], Heliou et al. [2017], even when $G$ is non atomic or with continuous action sets, see

Hadikhanloo et al. [2021], Perrin et al. [2020]). In the online optimization and learning community, fictitious play for two player games (or correlated FP for more than two players, where a player best responds to the correlated empirical past actions of the opponents) is precisely Follow The Leader (FTL), and Smooth Fictitious Play is Follow The Regularized Leader (FTRL) (see Shalev-Shwartz [2011], Cesa-Bianchi and Lugosi [2001], Belmega et al. [2018], Kwon et al. [2017], Giannou et al. [2021a,b]).

How can FP and SFP be extended to learn Nash equilibria in zero-sum and identical-interest games when the stage game is not fixed but changes with time (such as the stochastic game of Shapley [1953])?

The difficulty is that, in stochastic games (SG), a state variable evolves from a period to the next one with a probability depending on the current state and actions players take. Therefore, every player, when choosing its action at any given stage, must strike a balance between its instantaneous reward (determined by the current state and the current profile of actions) and its continuation future rewards, which depend in particular on the next state. Extending the result of Shapley [1953] in the zero-sum case, Fink [1964] proved that when there are finitely many players and actions (the framework of our paper), any discounted stochastic game (DSG) admits a stationary Nash equilibrium in mixed strategies (e.g. a decentralized randomized policy that depends only on the state variable).

Stationary Nash are the simplest possible equilibria that a dynamic game can have and it is our objective to construct independent learning algorithms that are provable to globally converge to them in finite zero-sum and identical-interest DSG. This is a hard problem because the payoff function of a player in a DSG is not own-payoff concave or quasi-concave when the players are restricted to their stationary strategies and thus, no gradient based method is guaranteed to converge, even to a local Nash equilibrium (Daskalakis et al. [2021]). Only recently some positive results have been obtained. That is, Leslie et al. [2020], Sayin et al. [2020], Baudin and Laraki [2022] combined the Fictitious Play behavioral strategy with a Q-learning like updating rule to design a family of decentralized rules such that the sequence of independent empirical distributions of the players converge to the set of stationary Nash equilibria in the two major classes of zero-sum and identical-interest ergodic DSG.

These latest results are based on ideas of the well-known and comparatively much better understood framework of Reinforcement Learning (RL). In this setting, there is only one player (a Markov Decision Process). The advent of efficient and model-free algorithms in this context such as Q-learning (Watkins [1989]) has had a lot of impacts on RL with numerous extensions, including offline Q-learning (Kumar et al. [2020]), double Q-learning (Hasselt [2010]) and a wide range of applications (Tai and Liu [2016], Kurin et al. [2020]). However, the convergence of Q-learning does not extend to the multiagent setting (see Wunder et al. [2010], Kianercy and Galstyan [2012]).

Unfortunately, the algorithms in Leslie et al. [2020] and Baudin and Laraki [2022] need the players to know precisely the model from the beginning (i.e., the transition and payoff functions). To avoid this drawback, our paper introduces a robust regularized version of their rules which combines Smooth FP, Q-learning-like rule (i.e., updates at every step an estimate of the continuation payoff) and empirical estimates of the unknown parameters. This leads us to a family of model-free independent learning algorithms, where the payoffs and the transitions are unknown to the players, and where the stage payoffs are imperfectly observed (e.g. randomly perturbed with a zero-mean noise). Our algorithms are proved to converge to the set of regularized stationary Nash equilibria in zero-sum and identical-interest ergodic DSG (those are a subset of stationary $\varepsilon$-Nash equilibria, and as $\varepsilon$ goes to zero (corresponding to a vanishing SFP), they refine the set of stationary Nash equilibria).

Another efficient class of algorithms in RL are the Projected Gradient Methods (PGM) and their stochastic version (Williams [1992], Sutton and Barto [2018]). Very recently, PGM and Stochastic PGM (SPGM) have been studied by Daskalakis et al. [2020] in zero-sum stochastic games to prove that when the players play independently a PGM (or a SPGM) with different time scales,[1] one can approach a best stationary equilibrium iterate[2] in a finite time $T(\varepsilon)$, whenever the stochastic game is episodic.[3] Leonardos et al. [2022] proved a similar convergence result in episodic identical-interest

---

[1]Daskalakis et al. [2020] proved that their convergence result fails if the players have the same time scale.

[2]I.e., there is $t \in \{1, ..., T(\varepsilon)\}$ such that (in expectation in case of Stochastic PGM), the players play an $\epsilon$ stationary Nash equilibrium at time $t$.

[3]A SG is episodic if, at every stage, there is a positive probability that the game stops.

DSG, as soon as all the players use a PGM or SPGM.[4] Those are extremely interesting and promising convergence results. Note however that they don't cover our framework because they assume the game to have a positive probability to stop (implying that their trajectories terminate almost surely in finite time) while in our ergodic setting, all the trajectories are infinite and the game never terminates. So, our results and those of Daskalakis et al. [2020], Leonardos et al. [2022] are non-comparable and complementary: they prove the existence of a best iterate approximation, we prove a time-average convergence, we use different algorithms and tools, and orthogonal assumptions (episodic vs ergodic).

The counterpart of FP in continuous time is best-response dynamics (Matsui [1992], Harris [1998]) and modern proofs for the convergence of SFP rely on the convergence of the continuous model combined with stochastic approximation techniques (see for instance Benaïm et al. [2005], Benaïm and Faure [2013], Hadikhanloo et al. [2021]). We follow a similar approach: we prove the convergence to regularized stationary Nash equilibria of an associated smooth continuous-time dynamics and, using some advanced stochastic approximation tools, deduce the convergence of our discrete-time rules. This is an important technical difference with Daskalakis et al. [2020], Leonardos et al. [2022] who can prove their results directly in discrete time, with an explicit finite convergence bound $T(\varepsilon)$. Our stochastic approximation methods do not provide us with a convergence rate, which is an open problem for SFP and FP even in the classical setting.

**Contributions**

- We introduce a family of independent learning algorithms in stochastic games that are model-free (unknown transitions and imperfect observation of the stage payoffs);
- We identify the corresponding smooth continuous-time dynamics, and show it globally converges to regularized stationary Nash equilibria in identical-interest and zero-sum DSG;
- From the continuous-time convergence, we deduce that in our two classes of DSG, the uncorrelated empirical frequencies of actions generated by our discrete-time algorithms almost surely globally converge to the set of regularized stationary Nash equilibria.

**Outline**    Section 2 gives the main definitions and notations of the paper. Section 3 introduces the smooth fictitious play procedure together with the continuation payoff updating rule. In order to prove the convergence of the discrete time algorithm, and also for its own sake, a smooth best-response dynamics with a simple continuation payoff up-dating rule dynamics is described in Section 4 followed with sketches of proofs of both our discrete-time and continuous-time systems. Section 5 describes in more details the related work. The appendix contains the detailed proofs. Appendix C explains an example and presents empirical results of our algorithm.

## 2  Preliminaries

Strategic situations where several agents interact, get rewards and modify an environment can be modeled as Stochastic Games (SG). In our settings, the number of players, actions and possible states are finite. We consider players that are interested in the so-called discounted reward on an infinite horizon, that is players strike a balance between instantaneous rewards and future ones.

**Stochastic games**    SG are tuples $G = (S, I, A, \{r_s^i\}_{i \in I, s \in S}, \{P_s\}_{s \in S})$ where $S$ is the state space (a finite set), $I$ is the finite set of players, $A^i$ is the finite action set of player $i$, $A := \Pi_{i \in I} A^i$ is the set of action profiles, $r_s^i(\cdot) : A \to \mathbb{R}$ is the stage reward of player $i$, and $P_s(\cdot) : A \to \Delta(S)$ is the transition probability map (where $\Delta(S)$ is the set of probability distributions on $S$).

**Main Restrictions**    We are interested in two classes of games: *zero-sum* stochastic games are two-players SG where $r_s^1 = -r_s^2$ for every $s$ and *team* stochastic games are such that the payoff

---

[4]Leonardos et al. [2022] proved their results in the larger class of potential stochastic games (they called Markov Potential Games). In such a class of games, the players are divided into two categories: either they do not influence the transition or they have the same payoff function up to a constant (see Holler [2020], Leonardos et al. [2022]). This last class is slightly larger than identical-interest SG is called *Team Stochastic Games* in Holler [2020]. All our convergence results extend to team stochastic games. The class of potential SG where the transitions are independent of the player's actions is easy to solve, just let each player uses a smooth FP myopically state per state independently.

functions of the players differ only by a constant (there is $r_s(\cdot) : A \to \mathbb{R}$ such that for every $i$ and $s$, $r_s^i(\cdot) = r_s(\cdot) + c^i$ for a constant $c^i$). A special case is *identical-interest* SG where all payoff functions are equal ($r_s^i(\cdot) = r_s(\cdot)$ for all $i$). An SG is *ergodic* if there exists $T \in \mathbb{N}$ such that for any sequence of actions of length $T$, the probability to reach any state $s'$ starting from any state $s$ is positive.

**Game Form**  A stochastic game is played as follows: starting from an initial state $s_0$, at every step $n \in \mathbb{N}$, every player $i$ choose an action $a_n^i$ given the history of play and the current state $s_n$. The next state $s_{n+1}$ is drawn from distribution $P_{s_n}(a_n)$.

**Discounted Payoffs**  We suppose that every player $i$ is interested in maximizing its discounted payoff, that is the expectancy of $\sum_{k \in \mathbb{N}} \delta^k r_{s_k}^i(a_k)$ where $\delta \in (0,1)$ is the discount factor.

**Strategies**  A behavioral strategy $\sigma^i$ for player $i$ is a mapping associating with each stage $n \in \mathbb{N}$, history $h_n \in (S \times A)^n$ and current state $s$, a mixed action $x_n^i = \sigma^i(n, h_n, s)$ in $\Delta(A^i)$. The behavioral strategy is pure if its image is always in $A^i$. A stationary strategy of player $i$ is the simplest of behavioral strategies. It depends only on the current state $s$ but not on the period $n$ nor on the past history $h_n$. As such, a stationary strategy can be identified with an element of $\Delta(A^i)^S$ (a mixed action per state interpreted as: whenever the state is $s$, $i$ plays randomly according to $x_s^i$). The set of stationary strategy profiles is $\Pi_{i \in I} \Delta(A^i)^S$. Set $X = \Pi_{i \in I} \Delta(A^i)$, so a stationary profile is an element of $X^S$. For $y^i \in (\Delta(A^i))^S$ and $x \in \Pi_{i \in I} \Delta(A^i)^S$, we denote by $(y^i, x^{-i})$ the stationary profile where $i$ changes its strategy from $x^i$ to $y^i$. A stationary profile $x \in X^S$ is a Nash equilibrium if and only if no player has a profitable behavioral deviation. Fink [1964] proved the existence of stationary Nash equilibria in every finite DSG and that it is sufficient to check pure stationary deviations.

**Regularizer**  In this paper, we are interested in exploratory algorithms, which may classically be generated by the use of some steep concave regularizer. This regularizer is added to the payoff functions $r_s^i$ and can be given several interpretations (see Fudenberg and Levine [1998], Hofbauer and Sandholm [2002] for details): it models the uncertainty of the payoff caused by the "trembling hand" of players or is a way to generate strict incentives to explore all the actions. Formally player $i$ maximizes a perturbation of its payoff function $r_s^i + \epsilon h^i$ under the following hypotheses:

$$h^i : X \to \mathbb{R}^+, \text{ strictly concave in } x^i, \ C^1 \text{ on the interior,}$$
$$\lim_{x^i \to \partial \Delta(A^i)} \| \nabla_{x^i} h^i(x) \| = +\infty \text{ and } \epsilon > 0 \tag{H1}$$

In this paper we study the convergence of some discrete and continuous time systems to regularized Stationary Nash equilibria, parameterized by the regularizers $(h^i)_{i \in I}$ and parameter $\epsilon$.

**Definition 2.1. Regularized Stationary Nash Equilibria** of a DSG are the stationary Nash equilibria of the DSG with the perturbed payoff functions $r_s^i + \epsilon h^i, i \in I$.

Similarly to stationary Nash equilibria, there exists at least one regularized stationary Nash equilibria Takahashi [1964].

*Remark.* In identical-interest (resp. zero-sum) SG, we suppose all players take the same (resp. the opposite) regularizer $h$ so as the regularized payoff functions remain identical. In this context, one can take for example a separable regularizer function which is equal to the sum (resp. the difference) of concave functions depending only on $x^i$.

*Remark.* If in a profile, all actions are $\beta$-optimal with respect to discounted payoff based on functions $r_s^i + \epsilon h^i$ (*i.e.,* deviations are at most $\beta$ profitable), then this profile is called a $\beta$-regularized equilibria.

## 3  Smooth Fictitious Play in Stochastic Games (Discrete-Time Algorithms)

**Fictitious Play in Repeated Games**  Introduced by Robinson [1951] and Brown [1951], fictitious play is a decentralized behavioral strategy to repeatedly play a fixed normal form game. At every step $n$, every player $i$ chooses an action $a_{n+1}^i$ that is a best response to the past empirical average action of other players: $a_{n+1}^i$ must maximize $r^i(\cdot, x_n^{-i})$ where $x_n^{-i} = (x_n^j)_{j \neq i}$ and $\forall j \in I, x_n^j = \frac{1}{n+1} \sum_{k=0}^n a_k^j$.

A famous variation of this procedure is smooth fictitious play (Fudenberg and Levine [1995]) where players choose their action according to a regularized payoff function. Formally, a player $i$ draws an action according to a distribution that maximizes $r^i\left(\cdot, x_n^{-i}\right) + \epsilon h^i\left(\cdot, x_n^{-i}\right)$ (with $h^i$ and $\epsilon$ defined in the previous section). An interesting property of such a procedure is that with suitable property on $\epsilon$, it has no *regret* (up to $\epsilon$), meaning that if played unilaterally by a player $i$, other players can not trick $i$ into using a suboptimal action. This is due to the randomness of the action choice: the distribution assigns positive probability to every action, so a player's behavior remains unpredictable.

In this section, we extend the smooth FP procedure to stochastic games. This builds upon the recent extension of FP in Leslie et al. [2020], Sayin et al. [2020], Baudin and Laraki [2022] (see Section 5 for more details). Indeed, it is not easy to derive the convergence to a regularized equilibrium by applying directly the definition of smooth fictitious play to the discounted stochastic game in which the players are restricted to play in stationary strategies, because the payoff function in this game is non-linear (nor is it concave, or quasi-concave) with respect to a player stationary strategy. To overcome this difficulty, and following the idea in Leslie et al. [2020], we update two sets of variables for every state: one concerns the uncorrelated empirical actions and the other is an estimate of the continuation payoffs (*i.e.,* payoffs that players can anticipate to achieve if that state is reached). These continuation payoffs are used as a parameter in an auxiliary game, often called the Shapley operator.

**Auxiliary game**  Following numerous authors including Shapley [1953], we define a so-called auxiliary game parameterized by a family of vector $u^i \in \mathbb{R}^S$ (one for every player $i$) and a state $s$. It is a one-shot game every player $i$ has the same action set as in $G$ but gets a one-time payoff of:

$$f_{s,u^i}^i\left(a_s\right) := (1 - \delta)r_s^i\left(a_s\right) + \delta \sum_{s' \in S} P_{ss'}(a_s)u_{s'}^i \tag{1}$$

Function $f_{s,u^i}^i$ is extended to mixed action profiles.

**Smooth Best-Response**  While playing the stochastic games, players maintain a set of continuation payoffs $u^i$ that are used to choose their actions. Given that the current state is $s$, for a player $i$, its action is drawn from the distribution that is the smooth best-response in the auxiliary game with respect to empirical action profile $x_s$, that is:

$$\mathrm{sbr}_{s,u^i}^i\left(x_s\right) := \arg\max_{y^i \in \Delta(A^i)} f_{s,u^i}^i\left(y^i, x_s^{-i}\right) + \epsilon h^i\left(y^i, x_s^{-i}\right) \tag{2}$$

This is well and uniquely-defined because of the strict concavity of $h^i$.

*Example.*  If the regularizer $h^i$ is taken to be the Shannon entropy, that is:

$$h^i\left(x_s\right) = -\sum_{j \in I} \sum_{a^j \in A^j} x_s^j(a^j) \log\left(x_s^j(a^j)\right)$$

Then the smooth best-response function is the logit function:

$$\mathrm{sbr}_{s,u^i}^i\left(x_s\right)\left(a^i\right) := \frac{\exp(\epsilon^{-1} f_{s,u^i}^i\left(a^i, x_s^{-i}\right))}{\sum_{b^i \in A^i} \exp(\epsilon^{-1} f_{s,u^i}^i\left(b^i, x_s^{-i}\right))}$$

See Mertikopoulos and Sandholm [2016] for other examples of regularizers.

**Smooth fictitious play with known transitions and deterministic payoff**  To extend smooth fictitious play to stochastic games, we use two sets of variables. The $x_{s,n}$ variable is the distribution of empirical actions for every state $s$ prior to step $n$. The other variable $u_{s,n+1}^i$ can be interpreted as the continuation payoffs at $s$ and is defined as the time average of the regularized payoffs up to step $n$. This leads to the following system:

$$\begin{cases} u_{s,n+1}^i = \dfrac{1}{n+1} \sum_{k=0}^{n} \left(f_{s,u_k^i}^i\left(x_{s,k}\right) + \epsilon h^i\left(x_{s,k}\right)\right) \\[2ex] x_{s,n} = \dfrac{1}{s_n^\sharp} \sum_{k=0}^{n} 1_{s=s_k} a_k \\[2ex] a_{n+1}^i \sim \mathrm{sbr}_{s,u_n^i}^i\left(x_{s,n}\right) \end{cases}$$

where $s_n^\sharp$ is the number of times $s$ was reached, that is $\sum_{k=0}^n 1_{s=s_k}$ and every action $a_k$ is embedded into the Euclidean space containing $\Delta\left(A^i\right)$. The interpretation is simple: at each stage $t$, each player best replies to the belief that the other players will play according to the past uncorrelated empirical distribution of actions, and that the future continuation payoffs are equal to the time average of the past estimated perturbed payoffs, calculated using the past empirical frequencies of actions.

This system can be generalized and rewritten in an incremental fashion. It is our first main system:

$$\begin{cases} u_{s,n+1}^i - u_{s,n}^i = \dfrac{\beta}{n+1}\left(f_{s,u_n^i}^i\left(x_{s,n}\right) + \epsilon h^i\left(x_{s,n}\right) - u_{s,n}^i\right) \\[2mm] x_{s,n+1} - x_{s,n} = \dfrac{1_{s=s_n}}{s_n^\sharp}\left(a_n^i - x_{s,n}\right) \\[2mm] a_{n+1}^i \sim \mathrm{sbr}_{s,u_n^i}^i\left(x_{s,n}\right) \end{cases} \tag{SFP}$$

where $\beta = 1$ corresponds to the equation above.

*Remark.* If there is only one state (e.g the classical repeated game sitting), this procedure is exactly standard smooth fictitious play since the smooth best-response does not depend on the value of $u_{s,n}^i$.

**Smooth fictitious play with unknown transitions and perturbed payoff**  We suppose now that payoff functions are not known, and that each stage payoff is observed with some zero-mean noise which follows a distribution that may depend on the history, the current state and actions taken by the other players. Therefore, at step $n$, player $i$ gets a random reward $R_i^n$ that is drawn according to a distribution determined by actions $a_n$ and current state $s_n$ whose expectancy is $r_{s_n}^i\left(a_n\right)$ and bounded variance conditionally on the history (see Appendix B for details). We also suppose that transitions are not known. Therefore, both transitions and expected payoff may be empirically estimated as follows:

$$\begin{cases} \hat{P}_{ss',n}\left(a\right) = \dfrac{\sum_{k=0}^n 1_{s_k=s \wedge a_k=a} 1_{s_{k+1}=s'}}{\sum_{k=0}^n 1_{s_k=s \wedge a_k=a}} \\[3mm] \hat{r}_{s,n}^i\left(a\right) = \dfrac{\sum_{k=0}^n 1_{s_k=s \wedge a_k=a} R_i^k}{\sum_{k=0}^n 1_{s_k=s \wedge a_k=a}} \end{cases} \tag{MFP.1}$$

Consequently, we define the estimated auxiliary payoff using these two estimators:

$$\hat{f}_{s,u^i}^i\left(x_s\right) := (1-\delta)\hat{r}_{s,n}^i\left(x_s\right) + \delta\sum_{s'\in S}\hat{P}_{ss',n}\left(x_s\right)u_{s'}^i \tag{MFP.2}$$

Now we define a model-free version of smooth fictitious play, similar to SFP but using estimators:

$$\begin{cases} u_{s,n+1}^i - u_{s,n}^i = \dfrac{\beta}{n+1}\left(\hat{f}_{s,u_n^i}^i\left(x_{s,n}\right) + \epsilon h^i\left(x_{s,n}\right) - u_{s,n}^i\right) \\[2mm] x_{s,n+1} - x_{s,n} = \dfrac{1_{s=s_n}}{s_n^\sharp}\left(a_n^i - x_{s,n}\right) \\[2mm] a_{n+1}^i \sim \mathrm{sbr}_{s,u_n^i}^i\left(x_{s,n}\right) \end{cases} \tag{MFP}$$

where $\mathrm{sbr}_{s,\cdot}^i$ is defined relatively to $\hat{f}_{s,\cdot}^i$.

**Theorem 3.1** (Convergence in identical-interest ergodic DSG). *In a identical-interest ergodic discounted stochastic game, if all players follow SFP (resp. MFP), their empirical actions $x_{s,n}$ converge almost surely to the set of regularized stationary Nash equilibria and their expected vector of continuation payoffs $u_{s,n}^i$ converges to the optimal continuation payoff of limiting equilibrium set.*

This theorem means that even if the trajectories of SFP (resp. MFP) cycle between several stationary equilibria, they all share the same optimal continuation payoff vector.

**Theorem 3.2** (Convergence in zero-sum ergodic DSG). *In a zero-sum ergodic discounted stochastic game, if all the players follow SFP (resp. MFP) with the same initial values, their empirical actions $x_{s,n}$ converge almost surely to the set of $D\beta$-regularized Nash equilibria (where $D > 0$ is a constant that only depends on $G$) and their expected vector of continuation payoffs $u_{s,n}^i$ converges to the corresponding continuation payoff.*

To prove these results, we are going to define in the next section an associated smooth best-response continuous time counterpart to our independent learning algorithms. The proof of Theorems 3.1 and 3.2 are sketched at the end of Section 4 and fully detailed in Appendix B.

*Remark* 1. Theorem 3.2 suggests that it is possible to use a doubling-trick mechanism to converge to the set of 0-regularized stationary Nash equilibria. Indeed, players can compute the duality gap (see Appendix A for a definition) and decide to reduce the update rate $\beta$ every time it is below a certain threshold. This is a standard trick to achieve no-regret in reinforcement learning.

*Remark* 2. Our rules suppose that each player observes the other player's past actions. This is an important difference with the algorithms developed in Daskalakis et al. [2020], Leonardos et al. [2022] where a player observes only its own actions. On the other hand, they do not prove a time-average convergence of their trajectories, while we do, but they prove a best iterate convergence that we do not prove.

## 4  Smooth Best-Response in Stochastic Games (Continuous-Time Dynamics)

**Continuous counterpart of discrete time systems**    The continuous-time counterpart of fictitious play is best-response dynamics of Matsui [1992], Harris [1998]. Together with the theory of stochastic approximations (Benaïm et al. [2005]), proofs of convergence of best-response are key ingredients to modern proofs of convergence of fictitious play and smooth FP in repeated games (Benaïm et al. [2005], Benaïm and Faure [2013]) and more recently in proving the convergence of FP in stochastic games (Sayin et al. [2020], Baudin and Laraki [2022]).

In this section, we define the continuous-time counterpart to our discrete-time algorithms. This leads us to a regularized version of the continuous-time best-response dynamics in Leslie et al. [2020], Baudin and Laraki [2022]. Similarly to the previous section, our continuous time system has two sets of variables $u_s^i(t)$ which represents the belief about the continuation payoffs and $x_s(t)$ which corresponds to belief about the players actions. Our main continuous-time system is:

$$\begin{cases} \dot{u}_s^i = \beta(t)\left(f^i_{s,u^i(t)}\left(x_s(t)\right) + \epsilon h^i\left(x_s(t)\right) - u_s^i(t)\right) \\ \dot{x}_s = \alpha_s(t)\left(\mathrm{sbr}^i_{s,u^i(t)}\left(x_s(t)\right) - x_s(t)\right) \\ \alpha_s(t) \in [\alpha_-, 1] \end{cases} \quad \text{(SBRD)}$$

*Remark* 3. In zero-sum or identical-interest games, if players have the same initial conditions, then we can omit the superscript $i$ in $u_s^i$ as they are equal.

**Update rates**    Profile $x_s(t)$ evolve towards the smooth best-response at a rate of $\alpha_s(t)$. Variable $\alpha_s(t)$ corresponds to the frequency at which a state $s$ is visited. The fact that it is bounded below by $\alpha_- > 0$ is a mathematically convenient way to exploit the ergodicity of the stochastic game.

*Remark* 4. Note that this is different from the model of continuous-time stochastic game outlined for instance by Neyman [2017]. In this paper, we are interested in using the theory of stochastic approximation, therefore discrete-time and continuous-time system are related through an exponential change of variable in time. As a consequence, we can consider that the state occupation is averaged, which is not the case in some other studies in stochastic games.

Regarding continuation payoffs, they evolve towards perturbed payoffs at a rate of $\beta(t)$, which is typically oblivious, that is only determined by time $t$ and state $s$ independently of the value of other variables. We make the following assumptions that guarantee that the update rates are not too small:

$$\beta(t) \geq 0 \text{ and } \beta \text{ is decreasing}$$
$$\int_0^\infty \beta(u)du = +\infty \quad \text{(H2)}$$

*Remark* 5. With $\beta(t) = \frac{1}{t+1}$, the system is close to the one outlined in Leslie et al. [2020]. In this case, continuation payoffs are updated with a *slower* timescale. Indeed, having two timescales may be useful to establish convergence or apply stochastic approximation theorems. In our paper, we do not make such an assumption, allowing the possibility that the two variables are on a single timescale.

**Model-Free System**   In order to also study the convergence of the model-free version of our procedure, we also define the smooth best-response dynamics when the model is progressively learned. The estimators are defined as follows for every pure profile $b \in A$:

$$
\begin{cases}
\dot{\hat{P}}_{ss'}(b) = \alpha_s(t) a(t)(b) \left( P_{ss'}(b) - \hat{P}_{ss'}(b)(t) \right) \\
\dot{\hat{r}}_s^i(b) = \alpha_s(t) a(t)(b) \left( r_s^i(b) - \hat{r}_s^i(b)(t) \right) \\
a^i(t) = \text{sbr}_{s,u^i(t)}^i (x_s(t))
\end{cases}
\tag{3}
$$

where $a(t) := \Pi_{i \in I} a^i(t)$ is the profile of selected actions and $a(t)(b)$ is the joint probability to select pure profile $b$ at time $t$.

Then, $\hat{f}_{s,\cdot}^i$ is defined as in (MFP.2) and we obtain a system MBRD by using estimators in SBRD.

**Solutions**   Systems SBRD and MBRD are differential inclusions. A general theory can be found in Aubin and Cellina [1984]. Here, the right-hand side of each system forms a closed, set-valued map (because there are several possible values for $\alpha_s(t)$) whose images are convex and uniformly bounded. Under these assumptions, it is known that such systems admit (typically non unique) solutions.

**Theorem 4.1.** *Under hypothesis H2, MBRD converges to the set of regularized Nash equilibria in identical-interest stochastic games. Furthermore, if $\beta^\star \geq \limsup_{t \to \infty} \beta(t)$, then MBRD converges to the set of $D\beta^\star$-regularized stationary Nash equilibria in zero-sum games, where $D$ is a positive constant that depends only on the game $G$. In particular if $\beta(t)$ goes to $0$ then MBRD converges to the set of regularized stationary Nash equilibria.*

The convergence of MBRD is helpful to characterize the limit set of discrete-time smooth fictitious play systems. Indeed, they are contained in the internally chain transitive sets (see Appendix B for a definition) of MBRD using the theory of stochastic approximations. Below, we sketch the proof of continuous-time Theorem 4.1. The discrete-time counterpart Theorem 3.1 and Theorem 3.2 can be deduced in a similar fashion as Baudin and Laraki [2022], although there are several technical subtleties. Complete proofs of these results are in Appendix A.

*Sketch of the proof of Theorem 4.1.*   The proofs for both class of games proceed quite differently: this is not surprising since there are no (to the best of our knowledge) unified convergence proof of even simple FP in potential and zero-sum games.

For identical-interest stochastic games, the key point is showing that the gap between (estimated) auxiliary payoffs $f_{s,u^i(t)}^i(x_s(t))$ and the auxiliary values $u_s^i(t)$ is narrowing. Technically, the difference is bounded below by an (absolutely) integrable function. Since it is the differential of $u_s^i(t)$ and that $u_s^i(t)$ it, which implies that $u_s^i(t)$ converges and then that $f_{s,u^i(t)}^i(x_s(t))$ converges and their limits are necessarily equal. Then, a study of the behavior of actions shows that they belong to the set of regularized equilibria, otherwise $f_{s,u^i(t)}^i(x_s(t))$ could not converge (based on Lipschitz properties of all these quantities).

Regarding zero-sum stochastic games, the first part of the proof studies a rather standard quantity called the duality gap. It goes to $0$ (or at least near $0$), which implies first that the value of the auxiliary game is mostly reached by the auxiliary payoffs $f_{s,u^i(t)}^i(x_s(t))$ and second, that the minmax strategies of the auxiliary game is learned by the players. Then, comparing relative speed of auxiliary values in all states leads to the convergence of values $u_s^i$ and of other variables.

$\square$

# 5   More on the Related Works

**Single-Player Markov Decision Process**   Watkins [1989] introduced Q-learning to solve Markov Decision Process with guarantees when the environment is stationary. The central idea is to estimate the so-called Q-values for every state-action pairs that represents the continuation payoff using the following scheme:

$$
Q(s,a) \leftarrow Q(s,a) + \beta(r_s(a) + \delta \max_{a'} Q(s',a') - Q(s,a))
\tag{4}
$$

where $s$ is the current state, $a$ the action played, $s'$ the next state. Q-values for other state-action pairs are left unchanged. This simple rule does not need any information about the transitions.

Since then, numerous extensions have been introduced, for instance double Q-learning (Hasselt [2010]) or Q-learning with deep learning (van Hasselt et al. [2016]). However, in multiagent systems, from the one player's point of view the environment is not stationary because it comprises other players that are also using some non stationary learning rule. Therefore, there is no guarantee and indeed it may not converge to the set of Nash equilibria (Wunder et al. [2010], Calvano et al. [2020]. Our paper is a way to overcome this difficulty with a different updating rule for estimating the continuation payoffs. This updating rule is closer to Expected Sarsa (Sutton and Barto [2018]) where the continuation values are moved towards the *expectation* of future payoffs.

**Nash Q-learning** Hu and Wellman [2003] is another way to learn Nash equilibria based on Q-learning, supposing that players have access to a Nash equilibrium of the auxiliary game parameterized by the Q-values. In our paper, we do not suppose that the players have access to such an oracle.

**Fictitious Play Algorithms in Stochastic Games** Brown [1951] and Robinson [1951] introduced fictitious play to learn the value of repeated fixed zero-sum game in a turn based fashion (see also Berger [2007]). It was later studied in the usual simultaneous updating by Fudenberg and Levine [1998] with numerous extensions, including smooth fictitious play. However, none of these procedures converge to equilibria in the repetition of the zero-sum normal form game $\Gamma$ induced by the stationary strategies of our DSG, mainly because the payoff functions of $\Gamma$, on a player own-strategy, are neither linear, nor concave Daskalakis et al. [2021] and so new idea are necessary to overcome this difficulty.

Vrieze and Tijs [1982] studied fictitious play in the context of varying stage games. However, similarly to the original fictitious play of Robinson and Robinson, it is designed as a way to compute the value of a zero-sum game, and it is not a behavioral strategy (i.e., there is no current state, the goal is to compute the minmax of a series of matrices). More recently, Leslie et al. [2020] proposed a best-response dynamics in continuous time which converges to stationary Nash equilibria in zero-sum DSG. The convergence was extended to discrete-time procedures and to identical-interest discounted stochastic games by Baudin and Laraki [2022]. These discrete-time and continuous-time systems are close to our systems, but they only work in model-based settings with no perturbation, whereas ours is suited for model-free settings and uses smooth best-responses. Sayin et al. [2020] introduced a fictitious play algorithm for zero-sum stochastic games in discrete time with a continuation updating rule close to Q-learning in a model-based or model-free setting: both players update a Q-table with an entry per state-action pair whereas our continuation payoff are only indexed by the state. Compared to our work, algorithm of Sayin et al. [2020] does not estimate the model explicitly, which may or may not be an advantage depending on the context. Furthermore, their action selection is not smooth–the selected action must maximize the Q-values with an $\epsilon$-greedy strategy in the model-free case. In contrast, our work uses smooth best-responses, whose main advantage is that in the non-stochastic setting, this is known to have no-regret properties, while any $\epsilon$-deterministic procedures are known to have regret. Future work could formalize this notion of regret in the context of stochastic games. Finally, Sayin et al. [2020] algorithm and proof have been established in the zero-sum case while we consider both zero-sum and identical-interest stochastic games. Sayin et al. [2022] very recently introduced another Fictitious Play with provable convergence in identical-interest stochastic games with single controller, which is a restriction we do not make in this paper.

**Smooth or Regularized Learning** Learning using regularizers is a widespread technique in machine learning, for instance with "follow the perturbed leader" (Cesa-Bianchi and Lugosi [2006]) or so-called stochastic fictitious play (Fudenberg and Levine [1995]). It makes it possible to have no-regret (Perchet [2014], Shalev-Shwartz [2011]) and from another point of view it is a substitute to the simpler $\epsilon$-greedy exploration scheme in reinforcement learning (Sutton and Barto [2018]).

**Optimistic Gradient Descent/Ascent** Wei et al. [2021] recently proposed another algorithm to learn in stochastic games. Similarly to ours, their method is rational in the sense that, if the opponent is playing some arbitrarily stationary strategy, then the agent learns to react optimally. However, this is restricted to zero-sum games and is a procedure quite different from ours, it uses projected gradient ascent/descent instead of fictitious-play like action selection.

**Local vs Global**   Our algorithms lead to all the players globally optimizing. Other algorithms such as COMA (Foerster et al. [2018]) lead to local optima, and so belong to another line of work.

## 6   Conclusion

We defined a number of decentralized continuous and discrete time systems with exploration to learn its own payoff function, the transition probability function, and which converge to regularized (and so $\varepsilon$-approximate) stationary Nash equilibria in discounted stochastic games. This is an area that has not been widely studied and as such, a number of questions remain to be answered–either regarding our systems or regarding other recent decentralized learning algorithms in stochastic games.

First, the theory of stochastic approximations gives no clue about the speed of convergence of the corresponding discrete-time algorithms. Moreover, even in continuous time, this question remains open even for potential games (Harris [1998]).Therefore, an interesting question in both the stochastic and the repeated game setting is, which guarantee FP and SFP have regarding the rate of convergence?

Second, and inspired by the convergence of vanishing fictitious play to equilibria in classical repeated games (see Benaïm and Faure [2013], Hadikhanloo et al. [2021]), it would be of interest to study a vanishing version of our algorithm with a parameter $\varepsilon(t)$ that goes to zero in a way that guarantees the convergence to (exact) stationary Nash equilibria of the DSG.

Third, convergence of FP or SFP in repeated game extends to infinite action games and to non-atomic games. It would be of interest to relax our finiteness assumption, at least for the action sets.

A challenging open problem is the design of independent learning algorithms that converge to stationary equilibria in ergodic zero-sum and identical-interest DSG without the knowledge of the other player's past actions. The Projected Gradient Methods of Daskalakis et al. [2020], Leonardos et al. [2022] have this minimal information property, but their results are proved only episodic SG.

Last but not least, obtaining a last-iterate convergence instead of a time average convergence (as in our paper) or best iterate convergence (as in Daskalakis et al. [2020], Leonardos et al. [2022]) is another interesting direction. However, it is known that there is no necessary last-iterate convergence with smooth fictitious play in the non-stochastic case (Giannou et al. [2021a]). Therefore, we can imagine that this would require a different algorithm but we emphasize that algorithms without last-iterate convergence such as ours are still of interest because of their behavioral, micro-economics and experimental foundations (Fudenberg and Levine [1998], Hofbauer and Sandholm [2002]).

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
