# OpenReview forum: "Smooth Fictitious Play in Stochastic Games with Perturbed Payoffs and Unknown Transitions"
_NeurIPS.cc/2022/Conference — NeurIPS 2022 Accept_

### Official Review · Reviewer_NBa1 · 2022-06-20

**Rating:** 7
**Confidence:** 3
**Soundness:** 3 good
**Presentation:** 3 good
**Contribution:** 3 good

**Summary:**

This work studies the convergence of Smoothed Fictitious Play dynamics for ergodic discounted stochastic games that are either zero sum or identical interest. The authors prove that the empirical actions of the players in both cases converge to the Nash equilibria of a regularized version of the original game. Importantly, the algorithms studied are both decentralized and model free. At a technical level, the authors first study the convergence properties of continuous dynamics and then use this result to prove asymptotic convergence for the discrete version of the dynamics as well.

**Questions:**

a) How does this work compare with [1] and potentially other work on ergodic discounted stochastic games? What are the advantages/disadvantages of each approach?
b) Are there particular reasons for studying the Smoothed Fictitious Play dynamics given that the related FTRL dynamics may have trouble with last iterate convergence to mixed Nash equilibria even in normal form games [Giannou et al., 2021a]?

**Strengths And Weaknesses:**

Regarding strengths, the paper is clearly written especially its technical components (Section 3 and 4). To the best of my knowledge the result and the key components of the proof techniques in this work are novel. The setting of decentralized and model free algorithms studied here is also very interesting with both theoretical and potentially practical impact.

Regarding weaknesses, there is a lack of comparisons with prior results in ergodic discounted stochastic games. For example, [1] provides convergence guarantees for the zero sum case while at the same time being model free. The advantage of [1] is that they get last iterate convergence, a concrete convergence rate without observing the opponent's actions. Their proof technique, relying on an actor-critic framework, seems also quite general compared to this work since it reduces a stochastic game to a sequence of slowly changing normal form games that each can be readily solved with algorithms like OGDA.

As it stands, I am not very confident about this work's future impact given existing results in alternative learning dynamics. I am leaning towards acceptance (Weak Accept, 6) but I am willing to increase my score if the authors address my questions below.

I have read the response of the authors and decided to increase my score to an (Accept, 7).

[1] Last-iterate Convergence of Decentralized Optimistic Gradient Descent/Ascent in Infinite-horizon Competitive Markov Games
Chen-Yu Wei, Chung-Wei Lee, Mengxiao Zhang, Haipeng Luo Proceedings of Thirty Fourth Conference on Learning Theory, PMLR 134:4259-4299, 2021.

---

> ### Author Response · Authors · 2022-08-02
> **Response to Reviewer NBa1**
>
> a) We were not aware of [1] and it is indeed related. It will be added to the related work section in a last version. Both papers prove convergence to stationary equilibria in zero-sum stochastic games and are rational in the sense that, if the opponent is playing some arbitrarily stationary strategy, then the decision maker learns to react optimally. There are however three main differences: first, in our paper, we proved time average convergence to regularized equilibria in identical-interest stochastic games in addition to zero-sum games while in [1] they study only zero-sum stochastic games (where they proved a stronger result: last iterate convergence). Second, their procedure is quite different from ours (it uses projected gradient ascent/descent instead of fictitious play) and while an empirical comparison would be interesting, we emphasize that smooth fictitious-play like procedures (and their continuous time counterpart) are interesting to study for their own sake: they are widely studied in game theory, economics and biology because they have behavioral and micro foundations (see for instance Fudenberg-Levine Book, Hofbauer-Sigmund book, or Sandholm book). Thus, if one believes that economic agent will behave according to simple behavioral rules (e.g. a logit better reply to the past observations) then it is important to study the long term behavior of such a system even if it does not offer the best convergence properties (last iterate). Finally, we prove the convergence in discrete time but also in continuous time, while in [1] only the dicrete time model is considered. It is important to note that evolutionary biology for example are mostly interested by continuous time systems (e.g. Hofbauer-Sigmund book).
>
> b) Regarding last-iterate convergence, as shown in Giannou et al. (a reference that will be added in a last version), it is indeed not guaranteed with our model (nor it is in all previous work we know about except [1]. That said, we believe that our proof methodology can be followed to obtain a last convergence result in an algorithm similar to ours but where one replaces the smooth best response equation with an online mirror descent equation. Moreover, as in a), we think that this kind of procedures are interesting to study for their own sake.
>
> > Fudenberg Drew, and David K. Levine. The theory of learning in games. MIT Press,  (1998).
>
> > Hofbauer Joseph and Karl Sigmund. Evolutionary Games and Population Dynamics. Cambridge University Press, 1998.
>
> > William Sandholm. Population Games and Evolutionary Dynamics. MIT Press, 2010.

---

> > ### Comment · Reviewer_NBa1 · 2022-08-07
> > **I have increased my score**
> >
> > I believe that the above discussion accurately summarizes the differences between this work and [1]. I decided to increased my score to (Accept,7), hoping that the techniques developed here will spur further research or have impact beyond AI/ML e.g., in economics or biology.

---

> > > ### Author Response · Authors · 2022-08-08
> > > **Thanks**
> > >
> > > We are glad that the reply is satisfactory and that it changes your mind positively.

---

### Official Review · Reviewer_6jk8 · 2022-07-04

**Rating:** 4
**Confidence:** 2
**Soundness:** 3 good
**Presentation:** 2 fair
**Contribution:** 2 fair

**Summary:**

This submission is about fictitious play in stochastic games. Fictitious play is a well-studied and natural dynamics in repeated games where players play best responses to the past average actions of the other players. A variation of this is smoothed fictitious play in which the payoff functions are regularized, which guarantees (under certain assumptions) that players have no regret. (Smoothed) fictitious play has also been studied for the class of stochastic games in which the payoffs depend on the current state of the system, which in turn depends on the actions of the players. Several results on the convergence of fictitious play in such games are known. This submission adds to these results.

Its first contribution is to introduce a smooth fictitious play dynamics for the case that the players do not know the transition properties of the stochastic game and observe only perturbed payoffs. It is then proven that in both identical-interest and zero-sum games, this dynamics converges almost surely to the set of regularized stationary Nash equilibria. These results are proven via a continuous version of the smooth fictitious play dynamics. For this continuous version, convergence results are proven that are then used to argue that also the discrete version converges.

**Questions:**

see "Strength and Weaknesses"

**Limitations:**

n.a.

**Strengths And Weaknesses:**

I frankly admit that this submission is not within my main area of expertise and I found the technical details not always easy to follow. The paper relies very much on previous work on (smooth) fictitious play and stochastic games. It is hard to follow without knowing the literature very well because important concepts are introduced only very briefly. My main suggestion concerning the writing would be to try to make the paper more accessible to non-experts. Apart from that the writing is mathematically precise, as far as I can tell. What did not become clear to me is the precise contribution of the submission. There is a lot of literature in the area and my impression is that both the discrete and continuous dynamics are very much inspired by and related to dynamics that have been studied before (for other games). Surely they have been adapted to stochastic games but (for me as a non-expert) this seems rather canonical. I cannot really say how novel the convergence proofs are and I would appreciate if the authors could comment on the technical challenges they had to tackle that are different from previous analyses in the literature.

---

> ### Author Response · Authors · 2022-08-01
> **Response to Reviewer 6jk8**
>
> We thank the reviewer for the time they devoted to our paper.
>
> On the technical challenges compared to the literature, we emphasize that this is the first work that we know which studies smooth best responses in the context of stochastic games. Furthermore, estimating the parameters of the model (payoffs and transitions) in the same time as the continuation payoffs and empirical actions is a technical challenge in the extent that it must be proven that these quantities converge on the same timescale. This was not in previous work to the best of our knowledge.
>
> To give a brief overview of the literature of decentralized learning in games, it was mostly focused in learning static equilibria of a game that is repeated infinitely, the main algorithms are fictitious play (Brown 1951, Robinson 1951), smooth fictitious play (Fudenberg and Levine 1998, Hofbauer and Sandholm 2002), (projected) gradient descent, online mirror decent (Beck and Teboulle 2003), etc. Those algorithms have been proved to converges to Nash or approximate equilibria when the normal form game which is repeated is a zero-sum, identical interest or a monotone game.
>
> Only recently (last 3-4 years) some papers started to explore decentralized learning algorithms when the game played changes with time (e.g. a stochastic game) and without solving it at every step (which leads to another line of work such as Nash Q-learning, Hu and Wellman 2003). Most of the recent results (about 4 papers) studied zero-sum stochastic games and two papers only studied identical interest stochastic games. The main difficulty is that if convergence holds, then each player must be solving a Bellman equation in his MDP (assuming the other players are playing stationary). Hence, the algorithm we need to build needs to mix the Q-learning mechanics with the equilibrium of interests properties that classical game theory algorithms achieve. So, if the tools/idea used share some similarities with the classical tools/idea (which is necessary because we are extending to stochastic games what was true in classical repeated games and MDPs), we are facing many difficulties: just building an algorithm which converges in the simulations was a challenge! For instance, Q-learning used independently by all players does not necessarily converge (Wunder et al. 2010).
>
>
> > Brown, George W. "Iterative solution of games by fictitious play". Activity analysis of production and allocation 13, n 1 (1951): 374‑76.
>
> > Robinson, Julia. "An Iterative Method of Solving a Game". The Annals of Mathematics 54, n 2 (september 1951): 296. https://doi.org/10.2307/1969530.
>
> > Hofbauer Josef and William H. Sandholm. "On the Global Convergence of Stochastic Fictitious Play". Econometrica 70, n 6 (2002): 2265‑94.
>
> > Fudenberg Drew, and David K. Levine (1998). The theory of learning in games. MIT Press.
>
> > Hu, Junling, and Michael P. Wellman. "Nash q-learning for general-sum stochastic games". The Journal of Machine Learning Research 4: 1039‑69.
>
> > A. Beck and M. Teboulle. Mirror descent and nonlinear projected subgradient methods for convex optimization. Operations Research Letters, 31(3):167–175, 2003
>
> > Wunder, Michael, Michael Littman, and Monica Babes. ‘Classes of Multiagent Q-Learning Dynamics with Epsilon-Greedy Exploration’. In Proceedings of the 27th International Conference on International Conference on Machine Learning, 1167–74. ICML’10. Madison, WI, USA: Omnipress, 2010.

---

### Official Review · Reviewer_gpaa · 2022-07-04

**Rating:** 6
**Confidence:** 5
**Soundness:** 3 good
**Presentation:** 2 fair
**Contribution:** 3 good

**Summary:**

This article introduces a smooth fictitious play procedure for stochastic games, providing convergence proofs in some classes of games (two-player zero-sum and identical interest discounted reward games). The procedure does not require the players to know the reward functions and transition probabilities in advance, instead combining the fictitious play with learning these game parameters. The players then use these estimates to estimate the standard auxiliary game payoffs and learn continuation payoffs, following recent approaches by Leslie et al (2020) and Sayin et a (2020). The proof technique is to use asynchronous stochastic approximation, relating the discrete-time learning algorithm to the continuous time mean field dynamics - results from Baudin and Laraki (2022) are used to support the connection, so much of the effort in this article is proving the convergence of the continuous time dynamics.

**Questions:**

What exactly is the contribution of the article, in the context of the most recent works (in particularly Sayin et al 2021)? The Sayin algorithm could be deployed whenever your one could but I suspect it would be slower since it uses information less efficiently. Can you provide any evidence to support this? Or any other reason why your contribution improves our capabilities?

p4, Def 2.1, you define a regularised stationary Nash equilibrium as the equilibrium of a modified game in which the payoffs are regularised. I have two issues with this. Firstly, r^i_s+eps.h^i is a function from mixed strategies to the real line, but not from pure strategies / actions (which also matters in the sentence before (H1)). So you are now in the realm of continuous action games only, but the sources you cite show existence only in discrete-action games if I remember correctly. Also, the fixed points you claim convergence to apply smooth best responses / regularisation to the continuation payoffs, not to the instantaneous rewards; the equilibria will be different I think. Please can you clarify these issues.

p4, definition of x_n^{-i}, the a_k^{-i} are actions, not unit vectors. Furthermore, ignoring this distinction hides the fact that there is a difference between (standard) fictitious play and joint strategy fictitious play. In standard FP the opponents are assumed to be independent, so the x_n{-i} needs to be a concatenation of different mixed strategy estimates for each player, whereas the joint strategy fictitious play will estimate a single distribution over all opponent joint actions. Please clarify.

Line 176, you really should give a reference here for the no regret claim.

Equations (1) and (2), you drift seamlessly from f being defined over actions to f being defined over mixed strategies. Three lines later you use i as a "free parameter" on the left hand side, and a "dummy parameter" (being summed over) on the right hand side. Both could be forgiven, but are sloppy. I almost didn't mention them, but the impression it leaves is colouring my opinion of other parts of the manuscript and you should be aware of the effect of this kind of sloppiness.

At the top of p6, when you move to the online version of SFP, you lose the fact that x has stochastic evolution (through sampling of a_{n+1}). I think this is not important, but should at least be noted.

Remark 2 on p6 should really discuss the Sayin et al (2021) NeurIPS paper.

Too much is trying to be implicitly described by the notation in equation (3) on p7. In particular the notation a(t)(b) is not explained at all. I managed to work it out, but please can you define it more carefully?

In the paragraph "Repeated games" on p8, earlier work is unfairly maligned. Any of the reinforcement learning processes that rely on stochastic approximation will of course work with noisy observation of the rewards. This is so obvious that none of the works make a big deal of it. But it is mentioned all the way back in Fudenberg and Levine's textbook, and of course holds in Borgers and Sarin, Leslie and Collins, Benaim and co-authors, Borkar, etc etc.

The related works section again omits to discuss the Sayin et al (2021) NeurIPS paper. Why? It is after all referenced in the Baudin and Laraki paper which this current article builds quite heavily upon. This is especially important since the Sayin et al paper resolves one of the open questions in the conclusion (convergence when the other player's actions cannot be observed).


p15, in the proof of Lemma A.6, it is unfortunate that the small scalar time unit is called h, given the central importance of the regulariser function h. Please can you use different letters for these two concepts?

p15, in equations (6) and (7), I think the final terms should be u instead of Gamma?

First line of p16. Given the density of the ideas here, it would be really helpful if you could provide more explanation of why the concavity of the regularised payoff means the second term is positive.

Lemma A.7 claims that the scalars, u_s(t) and Gamma_s(t), converge to a vector, u_infty. The notation needs to be a bit more careful.

I am confused by the proof of Lemma A.7. Please can you clarify? I think I believe the calculations. They show that the integral of \dot{u})_s is bounded below. It is then claimed that this implies that u_s converges to its lim sup (perhaps the formula for \dot{u} that depends of the (as-yet-unknown) Gamma_s(t) is also important?). However the function u_s(t) = sin(t) also satisfies that the integral of dot(u)_s is bounded, but does not converge to its lim sup. What am I missing here? I am sceptical about this result, because u and Gamma depend on each other, but the proof of each seems to ignore the presence of the other, which is why I am being picky about asking for help to complete my understanding of your proof. If I remember correctly, similar parts of the Leslie et al proof work because x(t) is, for sufficiently large t, close to being a Nash equilibrium of the auxiliary game, irrespective of the behaviour of the u's, but that has not yet been established here.

In the penultimate line of p19, are you sure the inequality is justified? The first condition of the lemma states that the absolute value of the quantity is geq xi, but it seems the line in question uses that it's leq xi.



**Limitations:**

I do not think there is any potential (negative) societal impact here. The limitations are discussed adequately.

**Strengths And Weaknesses:**

The article is, I believe, the first to address the convergence of a stochastic fictitious play procedure in stochastic games. The model-based nature of the dynamics (explicitly learning the reward functions, transition matrices, and observing opponent behaviour for fictitious play) is likely, in my opinion, to be the fastest method for convergence in the setting studied, where players do not know the game in advance but can observe the states and the actions of others.

On the other hand, there is an implication (albeit not an explicit statement) that previous works do not prove convergence except when the game parameters are known. This is not true - the article "Decentralized Q-learning in Zero-sum Markov Games" by Sayin et al at NeurIPS 2021 proves convergence in zero-sum games with even fewer assumptions - the players do not even need to observe the actions taken by their opponents.

There are also problems in the precision of the writing, in the correctness/completeness of the proof, and in the justification/motivation of the article - there are not even any simulations or other performance comparisons against alternative approaches.

---

> ### Author Response · Authors · 2022-08-01
> **Response to Reviewer gpaa**
>
> We thank the reviewer for the time devoted to our paper and address the questions below.
>
> Regarding the comparison with Sayin et al. 2021, We acknowledge that we should significantly extend, in a final version, the comparison with Sayin et al. 2021 with the following elements: our method is different from the one in Sayin et al. 2021 from three fundamental aspects. The first one is that in the model-based version, we have estimate of state values, whereas Sayin et al. algorithm is based on state-action values. In the model-free version of our algorithm, we estimate the model explicitly while Sayin et al algorithm does not, which may or may not be an advantage depending on the situation. The second aspect is that in Sayin et al. algorithm, the action choice is not smooth--it must be an action that maximizes the Q-values with an $\epsilon$-greedy strategy in the model-free case. In contrast, our work uses smooth best-responses. The main advantage of our approach is that in the non-stochastic setting, this is known to be a no-regret strategy, while any deterministic procedures or any "$\epsilon$-deterministic" (procedures deterministic at every step with probability $1-\epsilon$) are known to have a regret. Future work should formalize this notion of regret in the context of stochastic games. Furthermore, smooth best-response have the advantage that they give less weight to the worst action, whereas $\epsilon$-greedy strategies are random explorations, regardless of the expected payoff of actions. Finally (and importantly), Sayin et al. algorithm and proof was established only in the zero-sum case and its extension to identical-interest stochastic games is an open problem which is not obvious because the identical interest property (e.g. all players facing the same optimization problem) is not preserved by their algorithm.
>
> For the reference about the existence of regularized Nash equilibria, it should indeed be this paper:
>
> > M. Takahashi (1964) Equilibrium points of stochastic non-cooperative n-person games. Hiroshima Mathematical Journal Volume 28, Issue 1, Pages 95 - 99
>
> which deals with compact action sets. In fact, one can easily check that the fixed point argument of Fink (in the finite case) works exactly the same in our smooth best response model, except that we need to use only Brouwer's theorem instead of Kakutani, as the best reponse in our regularized model is unique.
>
> Throughout our paper, actions are seen as embedded in a Euclidean space where the set of pure actions is a basis. Therefore, $x^i_n$ is a member of the simplex $\Delta(A^i)$, so $x_n^{-i}$
> denotes a list of elements of the simplex, technically it is a member of
> $\prod_{j \neq i} \Delta(A^j)$. Hence, our paper is not about correlated fictitious play, but independent fictitious play because we compute the time average actions of the players separately and independently across the players.
>
> Regarding the paragraph "Repeated games" on page 8, we agree with the reviewer and it will be removed in the final version.
>
> Regarding (6) and (7) on p. 15, it should indeed be $u$ instead of $\Gamma$.
>
> Regarding Lemma A.7, this will be extended in a final version. The point being made is that a bounded function whose derivative is lower bounded by an (absolutely) integrable function converge. The absolute value of function $\sin$ does not have a bounded integral. More generally, Lemma~A.7 works because  $u_s$ and $\Gamma_s$ are "almost" increasing, the term that make them not increasing is precisely what we bound--and it is shown that it is negligible as for the convergence. Note that in contrast to Leslie et al. work, this is for identical-interest stochastic games. In zero-sum game, there is no reason for $u_s$ to be, even "almost", increasing. There is a parallel with proofs of fictitious play in the non-stochastic case (player playing repeatedly the same game): in identical-interest games, it is shown that $u$ is increasing, whereas in zero-sum games, it is shown that the duality gap is decreasing.
>
> Regarding page 19 and the proof of Lemma A.11, there is indeed a mistake in the order of inequalities: $\left|\frac{dv_s}{dt}\right|$ must be bounded above using $|f_{s_f, u_s}-u_{s_f}|$. However, it is fixed by also using this quantity in $\dot u_s$ on line 703, i.e. by bounding it by $\beta(t)\left(\frac{(1-\delta)\xi}{4}-|f_{s_f, u_s}-u_{s_f}|\right)$. Then, when $\frac{d u_s - v_s}{dt}$ is bounded above, the factor before $|f_{s_f, u_s}-u_{s_f}|$ is $-1+\delta$ and is negative, so it can be safely bounded using the hypothesis. This will be repaired in a final version.

---

> > ### Comment · Reviewer_gpaa · 2022-08-08
> > **Score increased**
> >
> > I'm going to increase my score to weak accept. Your response has convinced me of the correctness, but I really hope you significantly improve the precision in the writing of the final version.

---

> > > ### Author Response · Authors · 2022-08-08
> > > **Thanks**
> > >
> > > We are glad that you are satisfied by the reply and thank you for the score increase. We will do our best to be more precise in the final version following your suggestions.

---

### Official Review · Reviewer_v2hz · 2022-07-11

**Rating:** 8
**Confidence:** 5
**Soundness:** 4 excellent
**Presentation:** 3 good
**Contribution:** 4 excellent

**Summary:**

This paper studies decentralized learning in  the context of stochastic games  with perturbed payoffs and unknown transitions. The paper extends the approach developed in  the algorithms in Leslie et al. [2020] and Baudin and Laraki [2022], by perturbing players' payoffs introducing a regularizer term. In particular,  the paper develops a learning rule that combines  smooth fictitious play, Q-learning-like rule, and empirical estimates of the unknown parameters (transitions and rewards). Exploiting this clever mix of learning models,  the paper proposes a set of  model-free  independent learning algorithms, which are shown to converge in zero-sum games and identical-interest games.  The paper is well written. In particular, the paper clearly explains the problem under consideration and the contributions made by the authors.
I have some minor suggestions\questions.

1. The paper  studies learning in SGs considering two important classes of games (zero sum and identical-interests). In order to provide a general idea  of the learning algorithm, it would be nice to have one or two concrete applications to understand the relevance of the contributions for applied work. These examples can be discussed  as part of the description of the SG model.
2. In order to show how the learning process works, it would be useful to generate some numerical exercises. I understand that given time and space constraints this may not be feasible.
3. In line 196 there is an example for the smooth best response $sbr_{s, u^{i}}^{i}\left(x_{s}\right)\left(a^{i}\right)$. In particular, the paper discusses the case of the multinomial logit model. Besides this case, is it possible to provide another example?
\smallskip

Overall this is a very nice paper, which studies a very important problem. It is well executed and    provides a new class of learning algorithms to study stochastic games. My recommendation is to accept the paper.


**Questions:**


I have some minor suggestions\questions.

1. The paper  studies learning in SGs considering two important classes of games (zero sum and identical-interests). In order to provide a general idea  of the learning algorithm, it would be nice to have one or two concrete applications to understand the relevance of the contributions for applied work. These examples can be discussed  as part of the description of the SG model.
2. In order to show how the learning process works, it would be useful to generate some numerical exercises. I understand that given time and space constraints this may not be feasible.
3. In line 196 there is an example for the smooth best response $sbr_{s, u^{i}}^{i}\left(x_{s}\right)\left(a^{i}\right)$. In particular, the paper discusses the case of the multinomial logit model. Besides this case, is it possible to provide another example?



**Limitations:**

I do not see any limitations in the current paper.

**Strengths And Weaknesses:**

The main strengths are the relevance of the problem and the excellent execution of techniques to solve the problem

---

> ### Author Response · Authors · 2022-08-01
> **Response to Reviewer v2hz**
>
> We thank the reviewer for the time spent on our paper and for the helpful remarks. Below are responses to the questions:
>
>  1. and 2.We implemented our algorithm and ran simulations on several stochastic games examples. Because of the space constraints, both examples and simulations can be added to the appendix of a final version.
>
> .3. On the choices of regularizers, we will add a remark with this reference:
>
> > P. Mertikopoulos and W. H. Sandholm. Mathematics of Operations Research, vol. 41, no. 4, pp. 1297–1324, November 2016.
>
> where there are several choices/examples of entropy: Shannon's entropy (discusses our paper) but also log barriere, Tsallis's or Rényi's entropy.

---

> > ### Comment · Reviewer_v2hz · 2022-08-10
> > **Accept**
> >
> > I'm very pleased with the revision of the paper. I happy to confirm my acceptance of the paper.

---

### Official Review · Reviewer_5KL1 · 2022-07-11

**Rating:** 7
**Confidence:** 3
**Soundness:** 3 good
**Presentation:** 4 excellent
**Contribution:** 2 fair

**Summary:**

The authors present a novel algorithm that strategic users can employ to learn an almost Nash equilibrium in dynamic games, without the need of knowing the state transition probabilities and their own payoff functions. The algorithm is based on a regularized version of fictitious play.

**Questions:**

1) Please better explain what is the advantage of the suggested method with respect to the one in Sayin et al 2021. It seems to me that both methods are model free (i.e., agents do not need to know the transition probabilities nor the functional payoff form).

2) Please clarify why the suggested algorithm does not require a two time scale separation.

3) The provided method only guarantees time-average convergence. Is it possible to derive last iterate guarantees?

4) Could you better explain where is the randomness coming from in the argument around line 176-178?

5) The sketch of the proof of Theorem 4.1 is not entirely clear. For example, in line 299 why does the fact that the difference is bounded from below imply convergence? Also, I could not follow the intuition for the zero-sum case.


**Limitations:**

The limitations are well discussed in the conclusion section.

**Strengths And Weaknesses:**

Strengths

-The problem of learning in dynamic strategic settings is of great interest.
-The method suggested by the authors is novel.
-The paper is written in a very clear way.

Weaknesses

-Given previous works in the literature, the obtained result is not surprising. In fact, the idea of extending fictitious play to dynamic games has already been suggested in the literature (yet by suggesting a different extension) and the convergence proof provided by the authors (based on continuous limit and stochastic approximation theory) follows the same main steps.
-The suggested algorithm only guarantees convergence to an approximate/regularized equilibrium.
-There are no results on the convergence rate. Since the approximation in Theorem 3.2 depends on the step size beta, I suspect that to have small error, the algorithm would require very small step-sizes, implying a long convergence time.

---

> ### Author Response · Authors · 2022-08-01
> **Response to Reviewer 5KL1**
>
>
> We thank the reviewer for the time they devoted to our paper. Below are responses to the questions:
>
> 1. We acknowledge that we should significantly extend, in a final version, the comparison with Sayin et al. 2021 with the following elements: our method is different from the one in Sayin et al. 2021 from three fundamental aspects. The first one is that in the model-based version, we have estimate of state values, whereas Sayin et al. algorithm is based on state-action values. In the model-free version of our algorithm, we estimate the model explicitly while Sayin et al. algorithm does not, which may or may not be an advantage depending on the situation. The second aspect is that in Sayin et al. algorithm, the action choice is not smooth--it must be an action that maximizes the Q-values with an $\epsilon$-greedy strategy in the model-free case. In contrast, our work uses smooth best-responses. The main advantage of our approach is that in the non-stochastic setting, this is known to be a no-regret strategy, while any deterministic procedures or any "$\epsilon$-deterministic" (procedures deterministic at every step with probability $1-\epsilon$) are known to have a regret. Future work should formalize this notion of regret in the context of stochastic games. Furthermore, smooth best-response have the advantage that they give less weight to the worst action, whereas $\epsilon$-greedy strategies are random explorations, regardless of the expected payoff of actions. Finally (and importantly), Sayin et al. algorithm and proof was established only in the zero-sum case and its extension to identical-interest stochastic games is an open problem which is not obvious because the identical interest property (e.g. all players facing the same optimization problem) is not preserved by their algorithm.
>
> 2. Our algorithm does not need two-timescale separation for identical-interest stochastic games (a problem not considered in Sayin et al.) where it converges to exact regularized Nash equilibria. On the other hand, in zero-sum stochastic games, while it does not use two timescales, it only converges to the approximate set of regularized Nash equilibria. Therefore, to achieve an arbitrary precision of the equilibria in the zero-sum case, one should use a two-timescale mechanism. For the identical-interest case, the two timescales are not necessary because the change of actions does not impede the convergence of values (although it may slow it).
>
> 3. On last iterate convergence, it is an interesting research question for a future, but necessarily different algorithm). Actually, we know that last iterate convergence does not hold for smooth fictitious play even in the non-stochastic case when the players play repeatedly the same game, so we cannot expect it to hold in our case. On the other hand, Sayin et al. algorithm satisfies only time average convergence and most of the recent existing algorithms only proves a best iterate convergence and not a last iterate convergence, at least in identical-interest stochastic games. We still believe that our work is of interest because smooth fictitious play and in particular logit choice model have behavioral, micro-economics and experimental foundations (see for instance Fundeberg and Levine book, or Hofbauer and Sandholm article below) and it is important to understand the long term behavior of such models. We will expand more on this limitation if the paper is accepted.
>
>  >   Hofbauer, Josef, and William H. Sandholm. "On the Global Convergence of Stochastic Fictitious Play". Econometrica 70, n 6 (2002): 2265‑94.
>
> > Fudenberg, Drew, and David K. Levine. The theory of learning in games. MIT Press, 1998.
>
> 4. The smooth best-response is a distribution that is computed by each player. This player then choose an action randomly according to this distribution. Hence the randomness is comming from the player's random choices of actions.
> 5. The difference is bounded from below and is the derivative of $u_s$, therefore its derivative is bounded below by an integrable function. Furthermore, $u_s$ is bounded, so it converges. Regarding the zero-sum case, the definition of the duality gap may help. The main point is that the empirical actions are close to the min-max strategies in the auxiliary game, which then makes it possible for the continuation payoffs to converge as well. The proof sketches would be extended in a final version.
>
>
> Furthermore, we emphasize that the theory of stochastic approximations is not well-suited to give convergence rates, which explains why they are not in our paper. This is a limitation common to papers of this field. This is a very interesting research question but would need substantial work to be addressed.

---

> > ### Comment · Reviewer_5KL1 · 2022-08-09
> > **minor additional comments**
> >
> > The authors addressed my questions satisfactorily. I only have a few more suggestions for the final version:
> >
> > 1) I believe that for the model-based version also Sayin et all. 2021 only requires estimates for state values
> > 2) I believe that some recent follow-up works of Sayin et all. 2021 address, at least partially, the identical interest case (see e.g. Fictitious Play in Markov Games with Single Controller and Logit-Q Learning in Markov Games). It may be interesting to compare the proposed work to these extensions.

---

### Meta-Review · Area_Chair_y6YR · 2022-08-25

**Recommendation:** Accept
**Confidence:** Certain

**Metareview:**

This paper examines the convergence of stochastic fictitious play (SFP) in certain classes of discounted stochastic games - more specifically, two-player zero-sum and potential / common-interest games. The players are not assumed to know the game's reward functions and/or transition probabilities beforehand, and instead "learn" - or, rather, estimate - these aspects of the game as they go. These estimates are subsequently used as proxies for the players' continuation payoffs, in the spirit of previous work by Leslie et al (2020) and Sayin et a (2020). The authors' proofs rely on asynchronous stochastic approximation techniques, and they leverage recent results of Baudin and Laraki (2022) to derive (and study) the mean-field, continuous-time limit of the SFP process.

The reviewers appreciated the paper's technical contributions, and the authors addressed the reviewers' concerns satisfactorily during the discussion phase; as a result, a consensus was quickly reached for an "accept" decision. I concur with this assessment but, at the same time, I would urge the authors to pay particular attention to the comments of Reviewer gpaa regarding the positioning of earlier work and some issues with the precision (and clarity) of the mathematical writing. With this proviso, I am happy to recommend acceptance as well.

**Award:**

No

---

### Decision · Program_Chairs · 2022-09-14

Accept